# Rapid Nanopore Sequencing of Positive Blood Cultures Using Automated Benzyl-Alcohol Extraction Improves Time-Critical Sepsis Management

**DOI:** 10.3390/antibiotics14101001

**Published:** 2025-10-09

**Authors:** Chi-Sheng Tai, Hsing-Yi Chung, Tai-Han Lin, Chih-Kai Chang, Cherng-Lih Perng, Po-Shiuan Hsieh, Hung-Sheng Shang, Ming-Jr Jian

**Affiliations:** 1Graduate Institute of Medical Science, National Defense Medical University, Taipei City 11490, Taiwan; 811010005@mail.ndmctsgh.edu.tw; 2Division of Clinical Pathology, Department of Pathology, Tri-Service General Hospital, National Defense Medical University, Taipei City 11490, Taiwan; cindyft12@mail.ndmctsgh.edu.tw (H.-Y.C.); doc31756@mail.ndmctsgh.edu.tw (T.-H.L.); imabacteriaman@mail.ndmctsgh.edu.tw (C.-K.C.); pcl@mail.ndmctsgh.edu.tw (C.-L.P.); iamkeith@mail.ndmctsgh.edu.tw (H.-S.S.); 3Graduate Institute of Pathology and Parasitology, National Defense Medical University, Taipei City 11490, Taiwan

**Keywords:** pathogen detection, sepsis, benzyl alcohol extraction, automation, Oxford Nanopore

## Abstract

**Background/Objective**: Timely identification of bloodstream pathogens is critical for sepsis management; however, PCR inhibitors such as sodium polyanetholesulfonate (SPS) in blood culture broth compromise nucleic acid recovery and long read sequencing. We assessed whether coupling a benzyl alcohol SPS-removal step to the fully automated LabTurbo AIO extractor improves Oxford Nanopore-based pathogen detection. **Methods**: Thirteen positive blood culture broths were pre-treated with benzyl alcohol and divided: half volumes were purified on the LabTurbo AIO; paired aliquots underwent manual QIAamp extraction. DNA purity was evaluated by NanoDrop and Qubit. Barcoded libraries were sequenced on MinION R9.4.1 flow cells for 6 h. **Results**: Automated eluates showed a median A_260_/A_280_ of 1.92 and A_260_/A_230_ of 1.96, versus 1.80 and 1.48 for manual extracts. The automated workflow generated 1.69 × 10^6^ total reads compared with 3.9 × 10^5^ reads for manual extraction. The median N50 read length increased from 5.9 kb to 8.7 kb, and the median proportion of reads classified to species increased from 62% to 84%. The hands-on time was <5 min and the sample-to-answer turnaround was <8 h, compared with >9 h and 90 min for the manual protocol, respectively. **Conclusions**: Benzyl alcohol SPS removal integrated into the LabTurbo AIO extractor yielded purer, longer, and higher read counts, enhancing nanopore sequencing depth and accuracy while compressing diagnostic turnaround to a single working day. This represents a practical advance for rapid blood culture pathogen identification in critical care settings.

## 1. Introduction

Bloodstream infections (BSIs) are a major global health problem, accounting for an estimated 48.9 million cases and 11 million deaths in 2017 alone [1]. Every hour of delay in administering effective antibiotics after septic shock onset increases mortality, underscoring the need for rapid pathogen identification [2]. Conventional culture workflows require 24–72 h for species-level identification, with antimicrobial susceptibility testing adding at least 1 more day—driving interest in same-day molecular diagnostics.

Long-read sequencing with Oxford Nanopore Technologies (ONT) has emerged as a promising solution. Proof-of-concept studies have demonstrated that metagenomic ONT sequencing can identify bacterial pathogens and resistance genes directly from respiratory samples within 6 h [3] and positive blood culture broths in real time [4]. Prospective intensive-care trials report accurate species calls and genotype-based susceptibility predictions in <10 h [5], whereas unified clinical pipelines combining host DNA depletion, streamlined library preparation, and cloud-based analysis have reduced sample-to-answer time to ≈7 h [6]. Recent work has also highlighted the value of Nanopore sequencing for high-resolution antimicrobial-resistance (AMR) surveillance. Chung et al. generated near-complete assemblies for 98 *bovine Salmonella* isolates, revealing mobile AMR determinants that were invisible to short-read data [7]. Likewise, Shelenkov et al. resolved plasmid-mediated amikacin resistance in multidrug-resistant *Klebsiella pneumoniae* from patients with COVID-19 in the intensive care unit using a hybrid Nanopore/Illumina approach [8]. These studies underscore the capacity of ONT to deliver both taxonomic and resistome information within clinically actionable time frames.

The quality of extracted nucleic acids remains the principal bottleneck in genomic sequencing workflows. Sodium polyanetholesulfonate (SPS), an anticoagulant in blood culture bottles, co-purifies with DNA and potently inhibits PCR and other enzymatic reactions. Fredricks and Relman first showed that benzyl alcohol phase separation efficiently partitions SPS from the aqueous DNA phase and restores robust amplification [9]. Subsequent refinements incorporating a benzyl alcohol/guanidine phase step followed by silica-column purification further improved inhibitor removal [10]; however, these protocols are manual and operator-dependent.

Preserving high-molecular-weight (HMW) DNA is equally important. Harsh bead-beating markedly shortens read length, whereas enzymatic or magnetic-bead chemistries yield longer fragments suitable for ONT sequencing [11]. In urine metagenomics, an enzymatic lysis protocol has been shown to increase the median read length 2.1-fold and boost mapped reads > 11-fold over mechanical lysis [12]. A six-kit comparison across matrices identified an HMW magnetic-bead kit as being superior for long-read metagenomics [13]. Cross-biospecimen evaluations confirmed that extraction bias can skew multi-omics analyses [14], and ultra-deep ONT datasets from mock communities highlighted the fragility of long-read performance to DNA quality [15]. Beyond infectious-disease applications, metagenomic next-generation sequencing has improved the diagnosis of meningitis and encephalitis when conventional tests failed [16].

Automation offers a pragmatic path to reproducibility and scalability. Integrating automated extraction modules can cut hands-on time by >60% and deliver sequencing-ready DNA in <30 min without sacrificing yield or accuracy [17]. However, no study has combined benzyl alcohol SPS removal with a fully automated platform optimized for ONT sequencing of positive blood culture specimens.

To bridge this gap, we evaluated the LabTurbo AIO automated extraction system, which incorporates benzyl alcohol phase separation, against a reference manual silica-column protocol. We compared DNA purity and yield, ONT read-length distribution, pathogen-classification accuracy, and total turnaround time in clinical blood culture isolates. By uniting inhibitor-tolerant chemistry with walk-away automation, our study aims to deliver a practical same-day sequencing workflow for critical care bacteraemia diagnostics.

## 2. Results

### 2.1. DNA Purity After Benzyl-Alcohol Pretreatment and Automated Extraction

Spectrophotometric analysis confirmed effective SPS removal and carryover after benzyl alcohol phase separation. Excluding two low-concentration AIO extracts whose NanoDrop ratios were not reliable (clBC-09 and clBC-12, flagged “NR” in Table 1), the automated LabTurbo AIO workflow generated DNA with a median A_260_/A_280_ ratio of 1.90 (interquartile range [IQR]: 1.89–1.99) and a median A_260_/A_230_ ratio of 1.90 (IQR: 1.28–2.08). Conversely, manual QIAamp extraction produced median values of 2.17 and 2.40, respectively (*p* = 0.03 for A_260_/A_230_, Mann–Whitney U-test). In our series, Qubit concentration was undetectable in 2/13 AIO extracts (clBC-09 and clBC-12) but in 0/13 Qiagen extracts. Among the remaining libraries, AIO showed higher DNA concentrations (median 43.8 ng/µL; IQR 32.8–79.35) than Qiagen (9.66 ng/µL; IQR 7.77–26.9), indicating that low concentration was an infrequent event (2/13) rather than a systematic limitation of the automated workflow. As summarized in Table 1, the lower DNA yield observed in these two AIO samples, which was not seen in their manual counterparts, might be attributed to a low starting bacterial concentration in the aliquots processed. In such cases, a standardized automated protocol may be less efficient at recovering scarce nucleic acids compared to a manual procedure.

### 2.2. Automated Workflow Increases Read Count, Read Length, and Assembly Contiguity

Oxford Nanopore sequencing metrics for all 13 paired extracts are summarized in Table 2. The automated LabTurbo AIO (LabTurbo, New Taipei City, Taiwan) workflow generated a cumulative 1.73 × 10^6^ reads (median, 91,259 per sample, IQR: 14,483–138,154), whereas the manual Qiagen workflow produced 3.9 × 10^5^ reads in total (median 3216 per sample, IQR: 1042–36,490). Total bases followed the same trend, exceeding 1.37 × 10^9^ for the top-yielding AIO sample compared with just over 3.4 × 10^8^ for its manual counterpart.

Read-length statistics likewise favoured the automated protocol. The median of median read lengths across AIO extracts was 1264 bp (IQR: 459–2514 bp) versus 1058 bp (IQR: 373–1174 bp) for manual extracts. The N50 value—a proxy for assembly contiguity—reached 18.3 kb in the best AIO sample. Overall, the automated workflow delivered a median N50 of 8.7 kb (IQR: 4.3–12.7 kb) compared with 5.9 kb (IQR: 4.2–7.8 kb) for the manual method. This difference, however, was not statistically significant (*p* = 0.1824, Mann–Whitney U-test).

Notably, the median per-read quality score was slightly lower in LabTurbo AIO samples (median Q score 10, IQR: 9–11) compared to manual samples (median Q score 11, IQR: 10–11). Per-sample quality-score distributions are shown in Appendix A. Across 13 pairs, AIO libraries had a median Q score of 10 (IQR 9–11; 6/13 libraries ≤ 9; 4/13 ≥ 11), whereas Qiagen libraries had a median of 11 (IQR 10–11; 2/13 ≤ 9; 8/13 ≥ 11). Even with slightly lower Q scores (median 10 vs. 11), species-level classification and assembly contiguity (N50) favored AIO (Table 2); however, analyses that require single-nucleotide accuracy (e.g., SNP calling or high-resolution plasmid reconstruction) are more sensitive to per-base error and may require greater depth and/or consensus polishing. We did not benchmark variant calling in this study.

### 2.3. Increased Sequencing Output and Read-Length Metrics

Oxford Nanopore sequencing shows a clear read-length advantage when nucleic acids are prepared with the LabTurbo AIO. Across the 13 paired extracts, median read length increased from 934 bp (IQR: 432–1439 bp) with the manual Qiagen protocol to 1264 bp (IQR: 914–3169 bp) with the automated workflow (*p* = 0.003, Mann–Whitney U-test). Mean read length followed the same trend, rising from 1175 bp (median across samples) to 4518 bp (Figure 1A). The mean and median read length improved in 9 of 13 AIO samples (Figure 1), confirming that the longer fragments seen in Table 2 translated into a general trend of longer reads at the single-sample level. The N50 benefit and the >4-fold increase in the total read count are detailed separately in Table 2.

### 2.4. Improved Species-Level Classification Accuracy

Kraken2/WIMP analysis demonstrated higher taxonomic resolution when DNA was prepared with the LabTurbo AIO. Across the 13 paired datasets, the automated workflow produced a median of 84% for reads mapped at the species level (IQR: 78–87%), compared with 62% (IQR: 52–68%) for the manual Qiagen method (*p* = 0.002, two-tailed Mann–Whitney U-test). As illustrated in Figure 2, one of the thirteen AIO libraries contained ≥80% reads assigned to a single dominant pathogen, whereas no libraries prepared manually reached that threshold. In some cases, such as clBC_09, the dominant species represented a relatively low proportion of reads (~17%); however, it remained the most abundant taxon and matched the culture result and was therefore considered concordant. Human background reads were ≤5% of the total in 10 out of 13 libraries processed by the LabTurbo AIO method. Detailed taxonomic assignments, including dominant species percentages, human read fractions, and concordance with culture results, are provided in Appendix A. These data suggest that the combination of benzyl alcohol pretreatment and LabTurbo automation delivers cleaner, better-resolved metagenomic signals, thereby facilitating confident pathogen identification without the need for deep sequencing coverage.

### 2.5. End-to-End Workflow Time and Operational Efficiency

Figure 3 depicts the chronological workflow achieved with the automated protocol. Removal of PCR inhibitors (SPS) via benzyl alcohol phase separation required 30 min, with <5 min of hands-on time. DNA purification on the LabTurbo AIO instrument requires an additional 30 min of walk-away runtime, delivering eluates that proceed directly to library preparation. PCR barcoding and clean-up (≈75 min) are identical for both workflows, after which MinION sequencing runs for 6 h with real-time base-calling. Bioinformatic analysis was performed concurrently; hence, the entire sample-to-answer continuum was accomplished in under 8 h. Contrarily, the manual Qiagen method demands approximately 70 min of hands-on manipulation and 90 min total extraction time, extending the diagnostic pipeline to more than 9 h. The acceleration provided by automation was accompanied by substantive performance gains. Automated eluates display NanoDrop/Qubit (N/Q) ratios of median ≈ 1.0, indicating efficient recovery with minimal solvent carryover (Table 1). The higher molecular-weight DNA obtained yielded longer mean and median read lengths, with a median read-length N50 of 8.7 kb versus 5.9 kb for manually processed samples (Table 2). The automated workflow also generates roughly four-fold more reads, of which a larger fraction map unambiguously to the dominant pathogen (median 84% vs. 62%; Figure 2). Collectively, these improvements shorten the time to confident bacterial identification and preliminary resistance profiling to within a single working day, thereby enhancing the clinical utility of ONT-based blood culture diagnostics.

## 3. Discussion

The combination of benzyl alcohol phase separation with the LabTurbo AIO extractor produced a workflow that shortened the sample-to-report interval to <8 h (Figure 3). The automated workflow typically increased the read yield and read length, though some purity/quantity showed variability (Table 1); the fragment length (Figure 1 and Table 2) and species-level mapping (Figure 2) surpassed values obtained with a conventional silica-column protocol. These gains translated into deeper coverage and more confident taxonomic assignments without extending sequencing time or altering the standard PCR-barcoding kit procedure.

SPS has long been recognised as a potent inhibitor of enzymatic reactions in blood culture broths; Fredricks and Relman first showed that benzyl alcohol extraction efficiently partitions SPS into an organic phase, restoring robust amplification [9]. Here, we demonstrate that this inhibitor-tolerant chemistry can be coupled to a walk-away instrument, reducing hands-on time to <5 min per specimen and delivering sequencing-ready DNA in 30 min.

Our <8 h end-to-end timeline is comparable to the 7–9 h workflows achieved with host DNA depletion or real-time adaptive sampling [18]; however, the proposed approach uses neither depletion kits nor customised run scripts and therefore preserves sequencing yield and simplicity. The long-read performance also compares favourably with other positive-blood culture studies. Harris et al. reported a median N50 of 7.5 kb after enzymatic depletion [5], whereas the automated protocol here yielded a median N50 of 8.7 kb (Table 2). Together with a four-fold increase in the total reads, this improvement raised the median proportion of species-mapped reads from 62% (manual) to 84% (automated) (Figure 2), offering richer data for downstream resistance-gene and plasmid analyses. Per-sample Q-score distributions are shown in Appendix A. A critical finding of our study is the trade-off between sequencing yield and data quality. While the LabTurbo AIO workflow substantially increased the total number of reads and median read length, this came at the cost of a modest reduction in per-read quality (median Q10 vs. Q11 for manual). Although this ~1-point difference did not affect species-level identification or contiguity metrics in our dataset, base-level applications (e.g., SNP calling or high-resolution plasmid reconstruction) may require additional depth and/or consensus polishing.

Table 1 shows the NanoDrop/Qubit (N/Q) ratios for AIO samples ranged from 0.76 to 4.10. Ratios near 1.0 indicate good agreement between spectrophotometric and fluorometric quantification, while higher ratios suggest NanoDrop overestimation from contaminants. Most AIO samples clustered around ~1.0 (see Table 1), with two low-concentration outliers flagged in Section 2.1, confirming efficient recovery with minimal solvent carryover. This reliability obviates repeat extractions that prolong conventional workflows and may exhaust limited specimen volume. Longer reads and higher depth further enable accurate consensus polishing and rapid pathogen identification within a single working day—an outcome likely to improve empirical-therapy optimization in sepsis management.

This investigation has a few limitations. First, the study is dependent on a positive blood culture signal, limiting its application in early sepsis. Second, only 13 single-morphotype blood culture specimens from a single centre were analysed; validation in larger, multi-centre cohorts and polymicrobial samples is warranted. Third, while sequencing was performed in real-time, library preparation remained manual and contributed ≈75 min to the timeline; forthcoming transposase-based kits may shorten this step further. Fourth, this was a purely methodological comparison without clinical outcome data. Fifth, we also note potential biological heterogeneity: extraction yields may differ between Gram-positive and Gram-negative organisms due to cell-wall architecture; further protocol refinements such as enzymatic pre-lysis may benefit Gram-positive species. Sixth, economic evaluation was not conducted; while reduced labour costs may be offset by capital investment in low-throughput settings, high-throughput laboratories may achieve cost-effectiveness through decreased processing time and accelerated reporting. Seventh, antimicrobial-resistance gene detection and plasmid reconstruction were not systematically assessed, though critical for sepsis management. Eighth, external validation in larger, multi-centre cohorts is needed to assess reproducibility.

## 4. Materials and Methods

### 4.1. Clinical Specimens

All procedures were conducted following the Declaration of Helsinki and were approved by the Institutional Review Board of Tri-Service General Hospital (TSGHIRB No.: B202405085; approved on 24 May 2024). Informed consent was obtained from all participants.

From January to March 2025, we prospectively retrieved 13 consecutive blood culture bottles that signalled positive on a BacT/ALERT Virtuo instrument (bioMérieux, Marcy-l’Étoile, France) within 1 h of the alarm. Ten bottles were aerobic and three anaerobic. Gram-stain of each broth showed a single bacterial morphotype. This positive signal served as the time-zero (T0) starting point for the subsequent < 8 h diagnostic workflow evaluated in this study.

### 4.2. Benzyl Alcohol Phase-Separation Pretreatment

Each positive broth (500 µL) was mixed 1:1 with lysis buffer (5 M guanidine-hydrochloride, 100 mM Tris-HCl pH 8.0; Sigma-Aldrich, St Louis, MO, USA). Benzyl alcohol (600 µL, ≥99.5%; Merck, Darmstadt, Germany) was added, the tube vortexed for 10 s, and centrifuged at 20,000× *g* for 5 min at 25 °C in a Model 5424 R centrifuge (Eppendorf, Hamburg, Germany). The upper aqueous phase (≈200 µL) was transferred to a new tube and processed immediately. A molecular-grade water blank underwent the same steps as a negative control.

### 4.3. Nucleic Acid Extraction

#### 4.3.1. Nucleic Acid Automated Extraction

The aqueous phase was purified on the LabTurbo AIO platform (LabTurbo Biotech, New Taipei City, Taiwan) using the manufacturer’s “BC_DNA_200 µL” script. DNA was eluted in 50 µL of AE buffer pre-warmed to 56 °C. Instrument run time was 30 min and required <5 min hands-on.

#### 4.3.2. Nucleic Acid Manual Extraction

An aliquot of the same pre-treated lysate (200 µL) was processed with the QIAamp DNA Mini Kit (Qiagen, Hilden, Germany) following the “Blood and Body Fluid Spin” protocol, including a double AW2 wash and 1 min dry spin at 20,000× *g*. DNA was eluted in 50 µL of AE buffer.

### 4.4. DNA Quality Assessment

Purity ratios (260/280 and 260/230) were measured with a NanoDrop 2000c spectrophotometer (Thermo Fisher Scientific, Wilmington, DE, USA). Concentration was determined with a Qubit 4 fluorometer and dsDNA HS Assay Kit (Thermo Fisher Scientific). Only samples satisfying the purity criteria of A_260_/A_280_ ≈ 1.8–2.0 and A_260_/A_230_ > 1.8 proceeded to library preparation.

### 4.5. Library Preparation and Nanopore Sequencing

Up to 400 ng of DNA were barcoded with the PCR Barcoding Kit SQK-PBK004 (Oxford Nanopore Technologies [ONT], Oxford, UK) according to protocol PBK_9125_v109_revS_24Mar2024 [19]. Amplification used LongAmp Taq DNA polymerase (New England Biolabs, Ipswich, MA, USA) and 14 cycles of 95 °C, 30 s, 62 °C, 15 s, and 65 °C, 3 min. Barcoded libraries were pooled, loaded onto an R9.4.1 FLO-MIN106D flow cell, and sequenced for 6 h on a MinION Mk1B device. Raw signals were base-called with Guppy v6.5.7 in high-accuracy mode (Q ≥ 7), demultiplexed, and classified with EPI2ME WIMP (2025.3.2) and Kraken2 v2.1.2 against the MiniKraken2-8GB_202410 database.

### 4.6. Bioinformatic Metrics and Statistical Analysis

Read-length and yield statistics were generated with NanoStat v1.6.0 [20]. Purity ratios, DNA yields, total reads, and N50 read lengths from automated versus manual workflows were compared with a two-tailed Mann–Whitney U test in GraphPad Prism v10.0 (GraphPad Software, San Diego, CA, USA). Statistical significance was set at *p* < 0.05.

## 5. Conclusions

Despite these limitations, the data show that benzyl alcohol SPS removal coupled with LabTurbo AIO automation generally provides higher DNA yield and longer read fragments, facilitating same-day Oxford Nanopore–based pathogen identification with minimal operator input. Wider adoption of this workflow may shorten diagnostic time-to-therapy and improve clinical decision-making, pending outcome-linked studies.

## Figures and Tables

**Figure 1 antibiotics-14-01001-f001:**
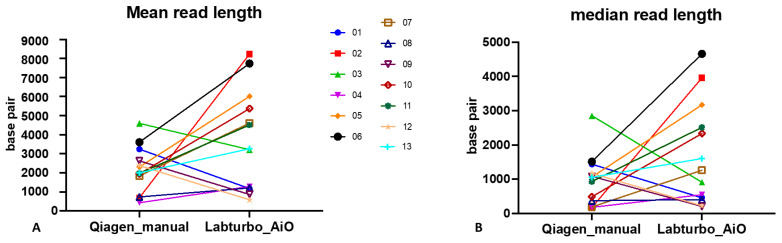
Automated LabTurbo AIO extraction yields longer ONT reads than manual Qiagen extraction. (**A**) Paired comparison of mean read length for each of the 13 positive blood culture specimens sequenced after manual Qiagen extraction or automated LabTurbo AIO extraction. (**B**) Paired comparison of the corresponding median read length for the same specimens.

**Figure 2 antibiotics-14-01001-f002:**
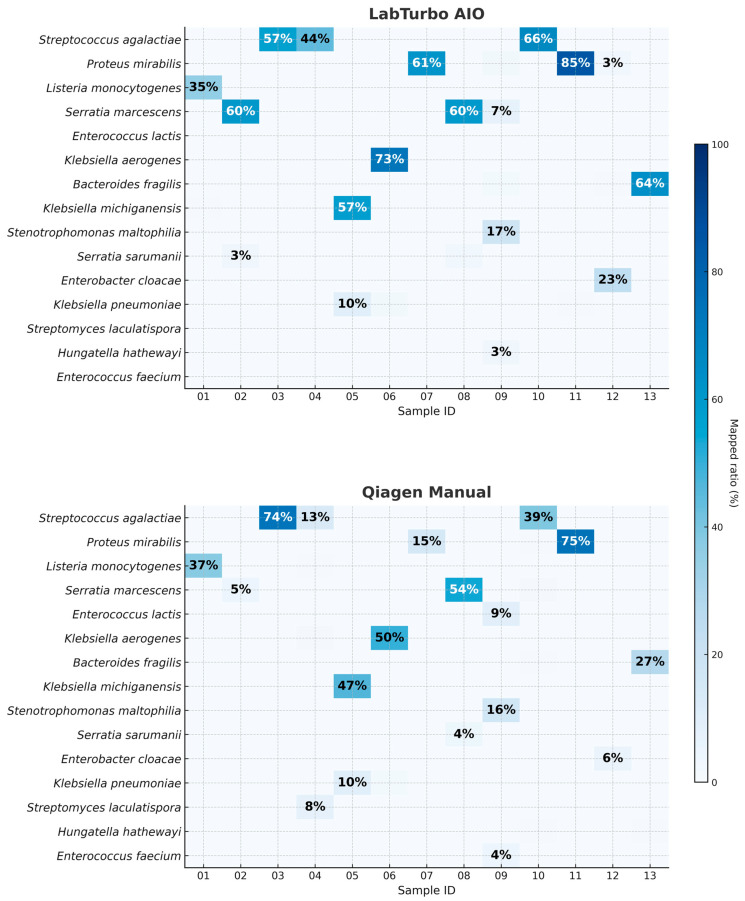
Species-level read-mapping percentages for the 13 positive blood culture specimens processed by automated and manual workflows. Each cell shows the proportion of total reads (%) classified by Kraken2/WIMP as the indicated species. The upper matrix (LabTurbo AIO) and lower matrix (Qiagen manual) are aligned by specimen ID (01–13). Values in cells represent a percentage of total reads mapped to the indicated species.

**Figure 3 antibiotics-14-01001-f003:**
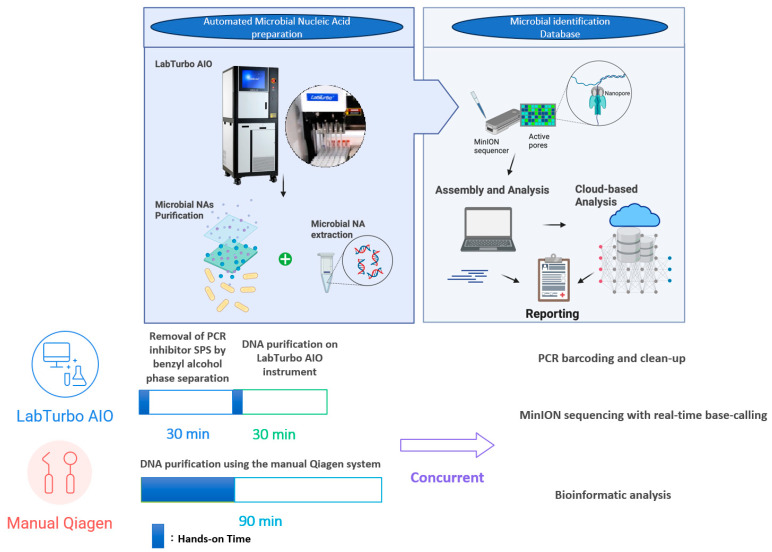
Comparative Workflow Timeline for Automated and Manual DNA Extraction.

**Table 1 antibiotics-14-01001-t001:** DNA purity and yield metrics for DNA extracted from 13 positive blood culture broths.

Nr	Method	NanoDrop Concentration (ng/µL)	260/280	260/230	Qubit (ng/µL)	N/Q Ratio **
Culture ID						
clBC_01 *L. monocytogenes*	*LabTurboAIO*	34.45	2.05	2.04	8.41	4.10
	*Qiagen*	43.20	2.46	1.13	6.37	6.78
clBC_02 *S. marcescens*	*LabTurboAIO*	67.10	2.06	1.83	66.80	1.00
	*Qiagen*	57.30	2.29	2.47	9.66	5.93
clBC_03 *S. agalactiae* (GBS)	*LabTurboAIO*	63.20	1.90	2.45	51.60	1.22
	*Qiagen*	47.20	2.16	3.98	13.70	3.45
clBC_04 *S. agalactiae* (GBS)	*LabTurboAIO*	261.40	2.05	2.32	91.90	2.84
	*Qiagen*	536.10	2.17	2.36	69.40	7.72
clBC_05 *K. michiganensis*	*LabTurboAIO*	49.80	1.94	1.18	40.30	1.24
	*Qiagen*	35.05	2.00	1.44	8.50	4.12
clBC_06 *K. aerogenes*	*LabTurboAIO*	66.40	1.87	0.66	25.30	2.62
	*Qiagen*	46.30	2.09	1.84	7.77	5.96
clBC_07 *P. mirabilis*	*LabTurboAIO*	60.15	1.90	1.37	42.20	1.43
	*Qiagen*	48.20	2.08	2.07	3.82	12.62
clBC_08 *S. marcescens*	*LabTurboAIO*	143.05	1.90	2.06	167.00	0.86
	*Qiagen*	88.80	2.06	2.58	81.10	1.09
clBC_09 * *S. maltophilia*	*LabTurboAIO*	12.75	NR	0.84	Low	NR
	*Qiagen*	25.40	2.41	1.74	8.19	3.10
clBC_10 *S. agalactiae* (GBS)	*LabTurboAIO*	51.85	1.88	1.90	43.80	1.18
	*Qiagen*	131.50	2.18	2.40	26.90	4.89
clBC_11 *P. mirabilis*	*LabTurboAIO*	22.95	1.91	1.19	19.60	1.17
	*Qiagen*	58.15	2.21	3.83	10.60	5.49
clBC_12 * *E. cloacae*	*LabTurboAIO*	9.75	NR	1.15	Low	NR
	*Qiagen*	16.65	2.40	3.10	4.37	3.81
clBC_13 *B. fragilis*	*LabTurboAIO*	179.15	1.87	2.09	236.00	0.76
	*Qiagen*	145.30	1.96	2.78	169.00	0.86

* clBC_09 and clBC_12 had concentrations below the instrument’s linear range; their aberrant A_260_/A_280_ values were therefore foot-noted as “not reliable” (NR) and excluded from group statistics. ** N/Q ratio: Ratio of concentration measured by NanoDrop (N) to that by Qubit (Q). A ratio closer to 1.0 indicates less contamination from RNA or residual solvents.

**Table 2 antibiotics-14-01001-t002:** Oxford Nanopore sequencing metrics for 13 positive blood culture specimens processed by automated LabTurbo AIO versus manual Qiagen extraction.

Nr	Method	No. of Reads	Total Base	Median Read Length	Mean Read Length	N50
Culture ID						
clBC_01 *L. monocytogenes*	*LabTurboAIO*	4111	4,833,591	459	1175	2986
	*Qiagen*	3216	10,429,773	1439	3243	7572
clBC_02 *S. marcescens*	*LabTurboAIO*	43,686	359,964,236	3959	8239	18,268
	*Qiagen*	413	285,148	178	690	7838
clBC_03 *S. agalactiae* (GBS)	*LabTurboAIO*	138,154	444,477,306	914	3217	10,145
	*Qiagen*	26,792	123,120,280	2854	4595	8761
clBC_04 *S. agalactiae* (GBS)	*LabTurboAIO*	73,297	94,102,975	550	1283	2857
	*Qiagen*	493	213,438	181	432	1425
clBC_05 *K. michiganensis*	*LabTurboAIO*	91,259	549,269,602	3169	6018	12,312
	*Qiagen*	2894	6,682,247	1067	2309	4997
clBC_06 *K. aerogenes*	*LabTurboAIO*	14,483	112,100,594	4660	7740	14,435
	*Qiagen*	1042	3,759,742	1513	3608	8944
clBC_07 *P. mirabilis*	*LabTurboAIO*	104,479	482,210,627	1264	4615	13,754
	*Qiagen*	637	1,171,550	194	1839	10,474
clBC_08 *S. marcescens*	*LabTurboAIO*	545,680	660,035,314	406	1209	4307
	*Qiagen*	77,605	57,104,905	373	735	1171
clBC_09 *S. maltophilia*	*LabTurboAIO*	684	591,111	205	864	5532
	*Qiagen*	84,725	227,474,249	1084	2629	5906
clBC_10 *S. agalactiae* (GBS)	*LabTurboAIO*	161,548	867,973,977	2334	5372	12,684
	*Qiagen*	2915	5,774,694	489	1946	6710
clBC_11 *P. mirabilis*	*LabTurboAIO*	135,079	610,332,467	2514	4518	8733
	*Qiagen*	36,490	73,806,550	934	1978	4185
clBC_12 *E. cloacae*	*LabTurboAIO*	129	75,002	227	581	2616
	*Qiagen*	23,181	57,514,746	1174	2382	4999
clBC_13 *B. fragilis*	*LabTurboAIO*	380,880	1,370,486,813	1605	3263	6864
	*Qiagen*	130,641	343,957,760	1058	1998	3876

## Data Availability

The datasets generated and analysed in this study are not publicly available due to legal and ethical restrictions imposed by the Tri-Service General Hospital Institutional Review Board (TSGHIRB No. B202405085). De-identified data can be made available from the corresponding author upon reasonable request and with prior approval from the TSGHIRB.

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
