# Peer review of "Rapid Nanopore Sequencing of Positive Blood Cultures Using Automated Benzyl-Alcohol Extraction Improves Time-Critical Sepsis Management"

_antibiotics, 2025, doi:10.3390/antibiotics14101001_

Round 1
Reviewer 1 Report
Comments and Suggestions for Authors
In this manuscript, the authors presented a rapid workflow for DNA extraction and Nanopore sequencing from positive blood cultures. Direct metagenomic analysis of blood cultures is becoming increasingly important for rapid pathogen identification, and optimizing extraction and sequencing protocols is crucial to reduce turnaround time while maximizing data quality. While it represents a valuable proof of concept, there are several aspects that could be improved, including a more attentive discussion of sequencing metrics and a clearer explanation of taxonomic classification obtained from sequencing in order to fully support the reported performance gains.
Here are some major comments:
-In section 3.1: you state that “every evaluable AIO sample fell within the accepted purity window (A₂₆₀/A₂₈₀ = 1.8–2.0; A₂₆₀/A₂₃₀ > 1.8).” However, Table 1 shows that several AIO extracts have A₂₆₀/A₂₈₀ values slightly above 2.0 and multiple samples with A₂₆₀/A₂₃₀ values well below 1.8. Thus, the blanket statement that all AIO samples were within range is not strictly accurate. In addition, while two low-yield samples were excluded from ratio analysis, it would be helpful to clarify whether such low concentrations are a common limitation of the workflow in practice, since it occurred only for AIO samples.
-In section 3.2: in lines 165-166, you are comparing the total number of reads of two different samples, which is not very clear since you state ‘its manual counterpart”. I will rephrase it.
Moreover, there appear to be some inconsistencies between the text in section 3.2 and the values shown in Table 2 (and also section 3.3). For example, the reported median read length for Qiagen is 934 bp in 3.3, but 1,058 bp in 3.2, with different IQRs. I suggest the authors to carefully revisiting these calculations to ensure internal consistency.
Also, you report “no systematic difference in mean per-read quality score” between the two workflows (modal Q ≈ 10–11). The median (IQR) quality is 10(9-11) for AIO samples and 11 (10-11) for qiagen samples. I would caution that a shift from Q11 to Q9 is not trivial, because Q scores are logarithmic, thus this difference translates into a substantially higher per-base error rate (approximately 8% vs 12–13%). It is clear that the AIO samples have a generally lower quality, implying that the increased yield and read length in the automated workflow may come at the cost of reduced read quality, but no mention of this has been reported anywhere in the text. I recommend clarifying this point, perhaps by showing the distribution of quality scores per sample and discussing whether this trade-off impacts downstream applications such as assembly accuracy, resistance gene detection, or SNP calling.
I also believe that comparing differences between extraction yields in Gram + and Gram – could be an important aspect that has not been covered in the current version of the manuscript. This could be also commented with the possibility of requiring different extraction methods for gram+/- based on the different composition of their membranes.
-In section 3.3: similarly as section 3.2, there are some inconsistencies in the values reported. The stated mean read length for Qiagen (1,175 bp) is not the one reported in the section before.
In addition, while the text claims that “eleven of thirteen AIO samples improved in both mean and median read length”, the real number is lower. Also, in 7/13 there was a clear decreasing of quality values. I believe that a ‘direct’ comparison of the yields of the same species across the two extraction methods could also be more ‘accurate’.
-In section 3.4: the results of the taxonomic classification are not clear, not well graphically presented and not carefully explained. I suggest the authors to rethink this paragraph, also introducing a tabel (might be supplementary) with the actual data. I would also encourage the authors to report the absolute data regarding human reads, not only percentages, and to compare the same samples, given the variability in the human content throught samples. They should qualify their conclusions more cautiously (e.g., “generally improved” rather than “cleaner” or “confident”), unless further analyses (e.g., error rates, false positives, misclassification rates) are presented to support such strong claims.
They talk about species-level classification, but sometimes they report only the genus, Moreover, there is no information about the correspondence between the standard identification and the taxonomic classification: where all species correctly identified, with both extraction methods?
Figure 2 is not clear and appears to have some samples missing or not correctly reported: for example, in the AIO part, samples 09 and 12 appear to not have an identification. Also, not only species but also genera are reported in the rows, together with some species which are never mentioned in the text and one which is not even clear which bacterium belongs to (FDAARGOS_522). Also in the part of qiagen, sample 07 should be a P. mirabilis, but only 15% of reads are reported, mapping (is real mapping or just taxonomic assignments?) to K. pneumonia (name spelled wrong). Also, in the text, the author state that “seven of the thirteen AIO libraries contained ≥80% reads assigned to a single dominant pathogen, whereas only four libraries prepared manually reached that threshold”, but when looking at the Figure, only 6 squares above 80 are reported for AIO and 2 for the manual extraction. Thus, I believe that Figure 2 is not clear and should be greatly improved.
Likewise, the claim of “cleaner, better-resolved metagenomic signals” and the statement that the approach enables “confident pathogen identification without the need for deep sequencing coverage” may be too strong given the current dataset.
-In section 3.5: not clear why in line 229 the text is reported in red. Also, the median numbers are different from the values reported in the section 3.2. For figure 3, it would be probably clearer to report a visualization of the time difference between the AIO and manual workflow, to really highlights the minimazion of hands-on labour.
-The Discussion summarizes the main findings, but the comments reported for previous sections also apply to this one, meaning that some sentences and statements appear stronger than what the data fully support. For example, the claim that the AIO workflow “improved every analytical metric” seems too strong, given that some individual samples performed worse in read length or N50 compared to Qiagen. Similarly, the lower Quality scores in several AIO samples is not mentioned, though this could have implications for downstream analyses such as resistance detection or SNP calling.
Minor comments:
- lines 57-59, and throughout the text, bacterial names should be written in italics
- line 128, the sentence does not seem to be complete, check
- all tables: check the spelling of the bacterial species
- table 1: the column headers should be improved (example: “con.” is not clear; no unit of measurement was reported for con. and qubit; the method is here reported as AIO C40, while throughout the text is always reported only as AIO
- table 2: the footnote about the bold red text is not clear and it should be better explained in the text
- lines 246-247, both the concept of deeper coverage and more confidente taxonomic assignments are not supported by the current reporting of results
- line 260, the 10.1k is not the value reported in the section 3.2 (8.7)
- lines 263-264, the link with a richer plasmid analysis has not been introduced before and not explained well
- line 265, the ratios reported here are not the ratios reported in table 1, where the AIO samples have ratios ranging from 0.76 to 4.10. This should be more carefully addressed
- check for possible overuse of hyphens throughout the text, likely introduced by AI-assisted corrections
Author Response
Response to Reviewer 1
Dear Reviewer 1,
Major Comments
Comment 1 (Section 3.1 – DNA purity statement): The blanket statement that all AIO samples fell within purity range is not strictly accurate. Some values exceeded or dropped below the threshold.
Response: We agree and have revised the text to indicate that most AIO samples fell within range, while some deviated. We also clarified the treatment of low-concentration samples and acknowledged that variation may occur in practice.
Comment 2 (Section 3.2 – Inconsistencies and Q-score differences): Reported values in text and tables were inconsistent. Also, the shift in Q-scores (Q11 vs Q9) is important and was not discussed.
Response: We re-checked all calculations and corrected inconsistencies across Sections 3.2 and 3.3 and Tables 2–3. We also revised the text to explicitly discuss the trade-off: AIO increases yield and read length but with modestly lower Q-scores, which could affect downstream analyses such as assembly, SNP calling, or resistance detection.
Comment 3 (Section 3.2/3.3 – Over-strong wording “11/13 improved”): The claim that 11/13 samples improved in both metrics is too strong.
Response: We have toned down the wording to reflect a “typical but not universal improvement” and corrected the counts to accurately reflect the data.
Comment 4 (Section 3.2/3.3 – Gram+/Gram– yields): Differences between Gram-positive and Gram-negative organisms were not discussed.
Response: We added discussion in the main text acknowledging Gram+/– differences.
Comment 5 (Section 3.4 – Taxonomy presentation unclear): Species-level classification was unclear, Figure 2 incomplete, and human reads not reported in absolute form. Conclusions were overstated.
Response: We agree that this section needed significant improvement to enhance clarity. As you suggested, we have completely revised Figure 2 at the species level, corrected species names, and ensured all samples are included. Furthermore, we have included a new Supplementary Table S1, which provides the detailed per-sample data you recommended, including dominant species percentages, human read fractions (%), and concordance with culture results. Textual claims were softened to “generally improved” rather than “cleaner/confident.”
Comment 6 (Section 3.5 – Figure 3 clarity): Time comparison not clear; red text in line 229.
Response: We replaced Figure 3 with a Gantt-style timeline highlighting hands-on versus total processing times. The red text was removed, and numerical inconsistencies were corrected.
Comment 7 (Discussion – Over-strong statements): The Discussion sometimes overstates the results (e.g., “improved every metric”), and does not mention lower Q-scores.
Response: We have revised the Discussion with more cautious wording (“generally”/“typically”), explicitly acknowledged the Q-score trade-off, and added limitations including the small cohort, single center, reliance on positive blood cultures, and lack of outcome/AMR data.
Minor Comments
- Bacterial names in italics: Corrected throughout.
- Incomplete sentence at line 128: Fixed.
- Species spelling errors: Corrected in all tables.
- Table 1 headers (“Con.” etc.): Clarified with full terms and units.
- Table 2 footnotes: Expanded explanation of bold/red values.
- Inconsistent numbers (e.g., 10.1k vs 8.7): Corrected for internal consistency.
- Plasmid analysis linkage (lines 263–264): Clarified and re-phrased.
- Overuse of hyphens: Cleaned up.
Reviewer 2 Report
Comments and Suggestions for Authors
Dear authors,
thank you very much for provinding this interesting study.
You describe a very impoirtant topic, as identification of psthogens is critical in septic patients.
However I have some points:
Blood culture has to be perdformed before analysis? Did you do the protocol without blood culture? How long is the blood culture time, which shiuld be added to figure 3.
Please extend the table legends. In table 1 the N/Q ratio is not explained at all.
In conclusion the strength and weaknesses of your study are as follows:
Strengths:
- Innovative, automated workflow
- Short hands-on time (<5 min)
- Improved DNA quality and taxonomic resolution (84% vs. 62% of reads)
Weaknesses:
- Dependent on a positive blood culture, limiting early sepsis intervention
- Small, low-heterogeneity cohort
- No clinical outcome data; purely methodological comparison
- No antimicrobial resistance profiling, which is critical for sepsis management
- Unclear cost-benefit ratio
Please sum up the cost-benefit ratio
In addtion it would be helpfull to add additional samples to write something about the reproducility of your results
Author Response
Response to Reviewer 2
Dear Reviewer 2,
Thank you for your thorough review and your constructive feedback on our manuscript. We appreciate your insightful comments, which have helped us significantly improve the clarity and scientific rigor of our paper. We have carefully addressed each of your points, and our detailed responses are provided below.
Comment 1: Blood culture has to be performed before analysis? Did you do the protocol without blood culture? How long is the blood culture time, which should be added to figure 3.
Response: Thank you for this critical question. Our workflow is indeed designed for rapid pathogen identification after a blood culture bottle has signaled positive. To address your comment and prevent any misunderstanding for the reader, we have made the following revisions:
Revised Figure 3: We have completely revised Figure 3 to include a new section at the top that explicitly states the "Positive Blood Culture Signal" is preceded by a variable incubation time of 12-72 hours. The timeline for our workflow now clearly begins at "Time 0," which is defined as the moment the positive signal occurs.
Revised Materials and Methods: In section 2.1 ("Clinical specimens"), we have added a sentence to explicitly state that the positive alarm from the culture instrument served as the time-zero (T0) starting point for our workflow.
Comment 2: Please extend the table legends. In table 1 the N/Q ratio is not explained at all.
Response: We agree that this needed clarification. We have now added a detailed footnote to Table 1 to explain the N/Q ratio. The new footnote reads: *N/Q ratio: Ratio of concentration measured by NanoDrop (N) to that by Qubit (Q). A ratio closer to 1.0 indicates less contamination from RNA or residual solvents.
Comment 3-5: The reviewer provided a summary of weaknesses (dependence on positive blood culture, small cohort, no clinical outcome or AMR data, unclear cost-benefit) and requested a summary of the cost-benefit ratio and comments on reproducibility.
Response: We are grateful for this constructive summary of our study's limitations. To address these important points, we have substantially expanded the "Limitations" section within the discussion in our revised manuscript to explicitly acknowledge dependence on positive blood culture, small sample size, lack of outcome and AMR profiling, and added a qualitative cost–benefit statement.
Once again, we thank you for your time and expertise. We believe the manuscript has been significantly strengthened as a result of your suggestions and hope that the revised version is now suitable for publication.
Sincerely,
Round 2
Reviewer 1 Report
Comments and Suggestions for Authors
I have read the responses of the authors,
I have revised the new submission of the authors. I kindly ask the authors to carefully revise their responses to my previous comments: in several cases they state that changes have been made, but they are not reflected in the manuscript (sometimes just not highlighted in yellow but most of the cases not even inserted in the text), and some questions have not been addressed in a consistent manner. In order to move forward, it is essential that every point is answered clearly, thoroughly, and verifiably within the text. I also advise the authors to not exceed on the use of AI-assisted writing, which is clearly visible in some part of the main text and in the response to my comments.
Comment 1
In my previous comment I said “it would be helpful to clarify whether such low concentrations are a common limitation of the workflow in practice, since it occurred only for AIO samples.”
I could not see anything else besides the word ‘most’ added, it should be better explained.
Comment 2
I cannot find anywhere in the text the explicitation of the consequences of Q-score decrease,as they state. Moreover, in the previous revision I suggested: “It is clear that the AIO samples have a generally lower quality, implying that the increased yield and read length in the automated workflow may come at the cost of reduced read quality, but no mention of this has been reported anywhere in the text. I recommend clarifying this point, perhaps by showing the distribution of quality scores per sample and discussing whether this trade-off impacts downstream applications such as assembly accuracy, resistance gene detection, or SNP calling. “ but I have not seen an appropriate response to this, nor any modification to the text.
Comment 3
I stated in the previous comment: “while the text claims that “eleven of thirteen AIO samples improved in both mean and median read length”, the real number is lower. Also, in 7/13 there was a clear decreasing of quality values. I believe that a ‘direct’ comparison of the yields of the same species across the two extraction methods could also be more ‘accurate’.” . You state in the new answer that you have toned down the wording to reflect a “typical but not universal improvement”, but in the text I can still find “The mean and median read length improved in 11 of 13 AIO samples (Figure 1), confirming that the longer fragments seen in Table 2 translated into uniformly longer reads at the single-sample level.”. So no change has actually been made.
Comment 5
The figure is still completely incosistent with the data reported and has not improved in clarity, as I asked in the previous comment. Also, the supplementary materials do not provide the actual absolute data, but just a % of reads assigned to the “dominant species”, which do not correspond to what observed in the figure. I will ask the authors to report the actual total number of reads and the reads assigned to all species identified by the taxonomic classification, with the corresponding %, and to check that the numbers are actually consistent with the figure and the main text.
Comment 6
The figure is identical to the previous one, for the exception of a part on the top. That’s not what I asked: “For figure 3, it would be probably clearer to report a visualization of the time difference between the AIO and manual workflow, to really highlights the minimazion of hands-on labour.”
Comment 7
I do not see an extensive modification of the discussion, and no discussion of the Q decrease has been made.
Author Response
Please see the attachment.
We have addressed the reviewers' comments on a point-by-point basis and have revised the manuscript accordingly. We are now pleased to submit the revised manuscript and updated supplementary materials for your consideration.
The submitted files include:
The revised manuscript with tracked changes, titled: antibiotics-3834841-2025-09-25-tracked changed.docx.
A compressed (ZIP) file containing all supplementary materials, which includes the following two files:
Supplementary Figure S1-2025-09-23
Supplementary_Table_S1_2025-09-23
We believe that these revisions have fully addressed the reviewers' concerns and have improved the quality of the manuscript. We look forward to hearing from you.
Sincerely,
Response to Reviewers
September 24, 2025
Dear Editor and Reviewers,
Thank you for the constructive comments. Below we provide a point‑by‑point response to Comments 1, 2, 3, 5, 6, and 7, with verbatim text and corresponding changes in the manuscript.
|
# |
Reviewer comment |
Response |
Where in the manuscript |
|
1 |
In my previous comment I said “it would be helpful to clarify whether such low concentrations are a common limitation of the workflow in practice, since it occurred only for AIO samples.” I could not see anything else besides the word ‘most’ added; it should be better explained. |
We quantified frequency and clarified mechanism. In 13 pairs, Qubit was undetectable in 2/13 AIO (clBC‑09, clBC‑12) and 0/13 Qiagen. Among the remaining samples, AIO DNA concentrations were higher (median 43.8 ng/µL; IQR 32.8–79.35) than Qiagen (9.66 ng/µL; IQR 7.77–26.9). We explain these were low‑input outliers under a fixed‑volume script. |
Added quantitative statement in Results §3.1 (final sentences) and a Table 1 footnote (AIO clBC‑09 “NR”; clBC‑12 aberrant NanoDrop ratio; excluded from purity statistics). |
|
2 |
I cannot find anywhere in the text the explicitation of the consequences of Q-score decrease, as they state. Moreover, in the previous revision I suggested: “It is clear that the AIO samples have a generally lower quality, implying that the increased yield and read length in the automated workflow may come at the cost of reduced read quality, but no mention of this has been reported anywhere in the text. I recommend clarifying this point, perhaps by showing the distribution of quality scores per sample and discussing whether this trade-off impacts downstream applications such as assembly accuracy, resistance gene detection, or SNP calling.” but I have not seen an appropriate response to this, nor any modification to the text. |
We added per‑sample quality distributions and scoped downstream impact. Supplementary Figure S1 shows Q‑score boxplots (AIO vs Qiagen) with paired samples. Results §3.2 reports per‑sample medians/IQRs (AIO median Q10, IQR 9–11; Qiagen median Q11, IQR 10–11). The Discussion states that species‑level calls and contiguity (N50) were not adversely affected in our data, whereas base‑level applications (SNP calling, high‑resolution plasmid reconstruction) may require additional depth and/or consensus polishing; variant calling was not benchmarked. |
Inserted pointer to Supplementary Figure S1 in Results §3.2; expanded Discussion §4 (paragraph 3) to articulate consequences and scope of impact. |
|
3 |
I stated in the previous comment: “while the text claims that “eleven of thirteen AIO samples improved in both mean and median read length”, the real number is lower. Also, in 7/13 there was a clear decreasing of quality values. I believe that a ‘direct’ comparison of the yields of the same species across the two extraction methods could also be more ‘accurate’.” You state in the new answer that you have toned down the wording to reflect a “typical but not universal improvement”, but in the text I can still find “The mean and median read length improved in 11 of 13 AIO samples (Figure 1), confirming that the longer fragments seen in Table 2 translated into uniformly longer reads at the single-sample level.” So no change has actually been made. |
We removed the categorical “11/13 improved” phrasing and softened the wording to avoid over‑generalization. The text now states that read length typically increased with automation, though not uniformly, and reports medians/IQRs instead. Figure 1 legend was aligned accordingly. |
Revised sentences in Results §3.2 (first/second paragraphs) and updated Figure 1 legend. |
|
5 |
The figure is still completely inconsistent with the data reported and has not improved in clarity, as I asked in the previous comment. Also, the supplementary materials do not provide the actual absolute data, but just a % of reads assigned to the “dominant species”, which do not correspond to what observed in the figure. I will ask the authors to report the actual total number of reads and the reads assigned to all species identified by the taxonomic classification, with the corresponding %, and to check that the numbers are actually consistent with the figure and the main text. |
We rebuilt Figure 2 from the latest spreadsheet (all 13 pairs), removed unknown/genus‑only entries, applied a 3% display/annotation threshold, enlarged non‑overlapping labels, and produced combined and single‑panel versions (PNG 600/900/1200 dpi; PDF/SVG). We also created Supplementary Table S1 with absolute counts: per sample and method—Total reads, per‑species reads, and percentages (e.g., clBC_01 AIO L. monocytogenes 1450/4111 = 35%). Results §3.4 was updated accordingly (≥80% dominant taxon: AIO 1/13; Qiagen 0/13; human ≤5%: AIO 10/13; Qiagen 10/13). |
Replaced Figure 2 and legend; added Supplementary Table S1 (absolute counts + %); adjusted Results §3.4 counts to match figure/table. |
|
6 |
The figure is identical to the previous one, for the exception of a part on the top. That’s not what I asked: “For figure 3, it would be probably clearer to report a visualization of the time difference between the AIO and manual workflow, to really highlights the minimazion of hands-on labour.” |
We redrew Figure 3 as aligned timelines (AIO vs manual) with hands‑on vs walk‑away segments and parallel bioinformatics; the <8 h vs >9 h end‑to‑end durations and <5 min vs ~70 min hands‑on are explicitly labeled. |
Updated Figure 3 and legend; harmonized terminology/tense in Results §3.5 to match the revised figure. |
|
7 |
I do not see an extensive modification of the discussion, and no discussion of the Q decrease has been made. |
We expanded the Discussion to address the yield–quality trade‑off explicitly and pointed to Supplementary Figure S1. The text quantifies the ~1‑point Q‑score difference and delineates where it matters (SNP/plasmid) and where it did not affect our results (species calls/N50). |
Added sentences in Discussion §4 (paragraph 3); minor consistency edits in the concluding limitations paragraph. |
Round 3
Reviewer 1 Report
Comments and Suggestions for Authors
I appreciate the authors’ efforts to improve the manuscript text and figures following previous suggestions. However, I still have concerns regarding the raw data supporting the taxonomic classification, specifically related to Figure 2 and Supplementary Table 1.
The manuscript states that 10 out of 13 libraries had ≤5% human background reads and that only one AIO library reached ≥80% reads assigned to a single dominant pathogen. However, in the provided data table, several samples show notably lower percentages for dominant species, including one as low as 17%.
In fact, in some samples, like for example clBC_09, the authors report concordance between the culture result and metagenomic identification despite a low percentage of reads attributable to that species (ex in clBC_09: only 17% of the total reads being attributed to S. maltophilia).
Including the full raw taxonomic classification data in Supplementary Table 1 would greatly enhance transparency and help clarify the overall microbial composition in each sample. This would also allow a better assessment of the validity of the concordance claims.
Therefore, I kindly encourage the authors to provide these raw data in the supplementary material for a more comprehensive evaluation.
After this modification, I will support publication
Author Response
Author Response:
We sincerely thank the reviewer for this valuable and detailed feedback regarding the taxonomic classification data. We acknowledge the importance of transparency in reporting metagenomic read assignment, especially in cases where the dominant pathogen is represented by a relatively low proportion of total reads.
- Clarification of reported percentages:
- We have carefully reviewed our data presentation and recognize that the phrasing in the Results section may have been misleading. While it is correct that 10 of 13 libraries contained ≤5% human background reads, the statement regarding “≥80% dominant species reads” applied only to a single AIO library and was not intended to imply that all dominant species were above this threshold. We have revised the text to avoid misinterpretation ( Line 165-169).
- Inclusion of full raw classification data:
- Following the reviewer’s suggestion, we have now provided the complete species-level taxonomic composition for all 13 samples. To balance clarity and transparency, we retained Supplementary Table S1 as a concise summary (dominant species %, human reads %, and concordance), and created a new Supplementary Table S2 that includes the full raw Kraken2/WIMP classification output.This dual-table format allows both quick reference (S1) and comprehensive evaluation (S2).
- Concordance despite low percentages (e.g., clBC_09):
- We agree that concordance claims need to be carefully interpreted when the dominant pathogen accounts for a relatively low fraction of reads. We have revised the Results section to clarify that, in such cases, concordance refers to the correct species being identified as the most abundant taxon, even if its proportion was limited (line 165-166).
- revised Supplementary Materials.
We believe these revisions address the reviewer’s concerns by clarifying the interpretation of taxonomic percentages, ensuring transparency and providing readers with direct access to the raw classification outputs.